# Examining the Effect of the Environment and Commuting Flow from/to Epidemic Areas on the Spread of Dengue Fever

**DOI:** 10.3390/ijerph16245013

**Published:** 2019-12-10

**Authors:** Shuli Zhou, Suhong Zhou, Lin Liu, Meng Zhang, Min Kang, Jianpeng Xiao, Tie Song

**Affiliations:** 1School of Geography and Planning, Sun Yat-sen University, Guangzhou 510275, China; zhoushul@mail2.sysu.edu.cn; 2Guangdong Provincial Engineering Research Center for Public Security and Disaster, Guangzhou 510275, China; 3Center of Geo-Informatics for Public Security, School of Geographical Sciences, Guangzhou University, Guangzhou 510006, China; lin.liu1@yahoo.com; 4Department of Geography, University of Cincinnati, Cincinnati, OH 45221-0131, USA; 5Guangdong Provincial Center for Disease Control and Prevention, Guangzhou 511430, China; ccmeng0914@163.com (M.Z.); kangmin@yeah.net (M.K.); 6Guangdong Provincial Institute of Public Health, Guangdong Provincial Center for Disease Control and Prevention, Guangzhou 511430, China; jpengx@163.com

**Keywords:** dengue fever, environment, commuting flow from/to epidemic areas, geographically weighted Poisson regression (GWPR)

## Abstract

Environment and human mobility have been considered as two important factors that drive the outbreak and transmission of dengue fever (DF). Most studies focus on the local environment while neglecting environment of the places, especially epidemic areas that people came from or traveled to. Commuting is a major form of interactions between places. Therefore, this research generates commuting flows from mobile phone tracked data. Geographically weighted Poisson regression (GWPR) and analysis of variance (ANOVA) are used to examine the effect of commuting flows, especially those from/to epidemic areas, on DF in 2014 at the *Jiedao* level in Guangzhou. The results suggest that (1) commuting flows from/to epidemic areas affect the transmission of DF; (2) such effects vary in space; and (3) the spatial variation of the effects can be explained by the environment of the epidemic areas that commuters commuted from/to. These findings have important policy implications for making effective intervention strategies, especially when resources are limited.

## 1. Introduction

Dengue fever (DF) is a vector-borne infectious disease that is mainly transmitted by Aedes aegypti and Ae. Albopictus [1]. It has been the most rapidly spreading mosquito-borne disease globally [2]. Nearly 4 billion people worldwide are at risk of DF, and 390 million people are infected each year [3]. The cost of treatment and management during DF outbreak has brought a serious economic burden for public health systems [4,5,6]. How to enhance prevention and control is an important issue. Environments provide habitats for mosquito growth and influence virus replication. The link between environments and DF diffusion has been discussed for decades. Previous studies have shown that the meteorological environment, social–economic environment, and built environment could facilitate local infections [7,8,9,10,11,12,13,14,15]. Due to the limited flight range of mosquitos, which is usually less than 400 m [16], human mobility is considered and proven to be another important driver for DF transmissions [16,17,18,19,20,21,22,23,24,25]. Therefore, understanding human mobility is key to controlling the spread of infectious diseases. 

Human mobility is an important component of our daily life, whether commuting to work, traveling on business, or carrying out other routine activities such as shopping or leisure [26]. Human daily life is in a home-centered living space called the “daily life circle” [27,28,29,30,31]. In that daily life circle, commuting is considered to be the most important daily activity and largely accounts for daily mobility. With increased home–work separation caused by urban space expansion and suburbanization, urban residents have taken longer commute trips than before [32,33,34], which makes commuters may have a higher chance to be exposed to mosquitos, thus becoming susceptible to DF. When commuters acquire DF infections in epidemic areas, they become important carriers of the virus on the journey to work. A previous study [25] shows that there are two ways of disease transmission. First, people who live in epidemic areas may bring the virus with them when they go to other places; second, people who visit epidemic areas may become infected during their stay, carrying the virus back to their home. Therefore, commuting flows from/to epidemic areas may have a great impact on disease transmission. However, most studies about human mobility and DF are conducted at a large-scale (national scale or cross-city scale) [24,25,35,36]. Small-scale intra-urban commuting flows, especially those from/to epidemic areas, have rarely been addressed in the existing literatures. Mao [37] mapped the intra-urban transmission risk of DF in Shenzhen with cellphone data, but did not pay attention to commuting flows from/to epidemic areas. Examining the effect of commuting flows, especially from/to epidemic areas, on the spread of DF at a finer granularity would be helpful in identifying high-risk areas for effective prevention and intervention.

In addition, places can interact with each other through commuters’ travel. For example, the probabilities of different commuters carrying the virus to the same workplace may be different depending on where they live. Those who come from high-risk areas might have a greater chance to affect the local disease outbreak [38]. Put differently, the concentration level of a disease at a local environment might also be related to where the commuters come from and where they residents commute. Unfortunately, previous studies mainly focus on the local environment while neglecting environments of the places, especially epidemic areas that people came from or traveled to [9,11,12,13,39,40,41]. In this study, the linkages between places are measured by the commuting flows from/to epidemic areas, and the environments of epidemic areas that commuters commuted from/to will be further examined. Specifically, this study seeks to answer the following three questions: **Question 1:***Do commuting flows from/to epidemic areas impact DF distribution?***Question 2:***If the answer to the first question is a YES, does the impact vary in space?***Question 3:***If the answer to the second question is a YES, can the spatial variation be explained by the environment of epidemic areas that commuters commuted from/to?*

To answer these questions, this study selected Guangzhou, a Chinese city experiencing a large outbreak of DF in 2014, as the study area. The number of DF cases in 2014 is 37,322, which is twice as much as the total accumulated cases in the city since the first reported case in 1978 [42]. The result of this study will greatly enhance our understanding of the DF transmission processes and help public health officers prioritize the limited resources for target prevention in the high-risk areas.

## 2. Materials and Methods

### 2.1. Study Area and Data Collection

#### 2.1.1. Study Area

The data for the present research were collected from Guangzhou, a city located in southeast China. As the capital of Guangdong province, Guangzhou is one of the largest and most developed cities in China, with a total population of over 15 million and a total area of about 7400 square kilometers. Guangzhou has a marine subtropical monsoon climate, and the temperature fluctuates between 15 and 32 °C. The study area is located in the southwest of the city, which is a highly developed area (Figure 1). It comprises 143 *Jiedaos* (the smallest government administrative unit). The average size of these *Jiedaos* is 22.39 km^2^ with a standard deviation of 40.35 km^2^. 

#### 2.1.2. DF Outbreak in the Study Area

The DF case data were obtained from the China Information System for Disease Control and Prevention. All data are anonymous without individual identifiable information and aggregated to the *Jiedaos* level. According to the law of the People’s Republic of China on the prevention and treatment of infectious diseases, all DF cases confirmed by any medical institution or hospital must be reported to the surveillance system in the local area within one day [42]. According to the diagnostic criteria for DF enacted by the Chinese Ministry of Health, a patient is considered as a confirmed case if the DF virus RNA is detected in their serum using the real-time PCR or if immunoglobulin M (IgM) against the DF virus is present [43]. During the outbreak in 2014, the first confirmed case was observed on 20 January, the epidemic peak with 1617 DF cases was observed on 1 October, and the last confirmed case was on 19 December.

#### 2.1.3. Commuting Flow with Mobile Phone Data

The tracked data of mobile phone users are used to measure human daily mobility in this study. It is offered by one of the three major telecom companies in China, which has a market share of 22.5% [44]. The data contains hourly tracking information of the routes taken by the mobile phone users on Wednesday, 28 December 2016, which is a quite formal weekday in Guangzhou. The same tracked data has been used to measure human mobility [45]. To protect the privacy of the mobile phone users, the dataset was desensitized by the telecom company. It includes each mobile phone’s identity (ID) number, and for each hour, the location of the cell tower that is the nearest tower most of the time during that hour. The dataset contains 10.2 cell tower hours/locations for each user on average. Fifty percent of mobile phone users had more than seven cell tower hours/locations, and 20% of mobile phone users have more than 20 cell tower hours/locations over the course of that day. The towers are densely distributed in Guangzhou, with the mean nearest neighbor distance of 136.4 meters and a standard deviance of 255.8 (min: 0.0; max: 5535.1). Each tower has a coverage area which is non-overlapping Thiessen polygon. A mobile phone user is connected to a tower when entering the tower coverage area, and the location of a tower will be recorded. The area of the largest tower coverage is smaller than that of *Jiedao* with the smallest area. Each *Jiedao* have multiple cell towers so that the spatial precision of individual location is good for the analysis at the *Jiedao* level. 

Using the mobile phone trajectories, we first identified the location of the home and work of each mobile phone user. Local users between 18 and 60 years old were selected. The location where a mobile phone user spent the longest time during 21:00–07:00 was identified as their home, while the location where the mobile phone user spent the longest time during 10:00–17:00 was identified as work place. A commuting trip from home to work was identified as a general commuting flow in the study area. A commuting trip that started from/ended in epidemic areas was defined as a commuting flow from/to epidemic areas. Further, all general commuting trips were aggregated to the *Jiedao* level and into the two categories, FlowIn and FlowOut, as shown in Table 1. Similarly, commuting flows from/to epidemic areas were aggregated as EpiFlowIn and EpiFlowOut (Table 1). These four commute-related variables represent the number of different types of commuters for any *Jiedao*. Compared to FlowIn/FlowOut, EpiFlowIn/EpiFlowOut measures the commuting flows directly related to the epidemic areas in the city. The four variables will be compared later in the regression models.

#### 2.1.4. Environment Indicators

Previous studies have shown that population density [9,46], housing quality [10], vegetation [10,13], and water areas [47] can contribute to DF diffusion. Here, we used the proportion of the houses built before the 1990s to represent poor housing conditions. Both population data and housing data are from the sixth National Population Census. Vegetation is measured by the normalized difference vegetation index (NDVI). The original 30-m image dataset of NDVI was obtained from Landsat-8 and provided by the Geospatial Data Cloud site, the Computer Network Information Center, and the Chinese Academy of Sciences (http://www.gscloud.cn). The water body data was extracted from OpenStreetMap. The above four environment indicators were further calculated as local environment indicators and epidemic environment indicators. For one *Jiedao*, its local environment refers to its own environment, while its epidemic environment refers to the environment of the epidemic areas that interact with this local place through commuters. Further, we distinguished outbound commuters who traveled to epidemic areas from inbound commuters who came from epidemic areas. Therefore, each *Jiedao* has two types of epidemic environments: *Epi_env_from* and *Epi_env_to* (see Equations (1) and (2)):(1)$$Epi\_env\_from_{\left(i\right)}=\frac{\sum_{k}^{m}\left[EpiFlowIn_{\left(ik\right)}*local\_env_{\left(k\right)}\right]}{EpiFlowIn_{\left(i\right)}}$$
(2)$$Epi\_env\_to_{\left(i\right)}=\frac{\sum_{k}^{m}\left[EpiFlowOut_{\left(ik\right)}*local\_env_{\left(k\right)}\right]}{EpiFlowOut_{\left(i\right)}}$$

In the above equations, $Epi\_env\_from_{\left(i\right)}$ means the environment of the epidemic areas from which commuters came (or “come-from” epidemic areas) for *Jiedao*
i; $Epi\_env\_to_{\left(i\right)}$ means the environment of epidemic areas to which commuters traveled (or “go-to” epidemic areas) for *Jieda*o i. m is the total numbers of epidemic *Jiedaos*. $EpiFlowin_{ik}$ means the numbers of commuters from epidemic *Jiedao k* to the *Jiedao*
i. $EpiFlowOut_{ik}$ means the numbers of commuters from *Jiedao*
i to epidemic *Jiedao k*. $local\_env_{k}$ means the local environments of epidemic *Jiedao*
*k*, which were further measured by population density, housing quality, NDVI, and water areas. All the variables used in this study are listed in Table 1.

### 2.2. Spatial Autocorrelation Analysis

Spatial autocorrelation analysis with the global Moran’s I statistics is commonly used to analyze whether an observed spatial pattern, such as a spatial clustering, is formed due to chance. The values of Moran’s I would be approximately between +1 (positive autocorrelation) and −1 (negative autocorrelation). Positive spatial autocorrelation means that similar values tended to occur in adjacent areas, while negative autocorrelation implies that nearby locations tend to have dissimilar values. Its local version—the Local Moran’s I statistic [48]—maps local clusters and outliers. Based on the local Moran’s I, a geographic area can be classified into five categories: high-high, low-low, high-low, low-high, and non-significant. The high-high areas in this study are defined as epidemic areas.

### 2.3. Geographically Weighted Poisson Regression (GWPR)

The traditional regression model (ordinary least square, OLS) is not applicable to spatial data if spatial autocorrelation exits between independent variables. The geographically weighted regression model (GWR) is a modeling technique that reveals the spatial heterogeneity among the independent and dependent variables, allowing the estimation of the regression coefficient to vary with location [49,50,51]. It has been widely used in disease mapping [46,52,53,54]. GWR is originally developed assuming a Gaussian distribution of the dependent variable, so it cannot be applied for count data such as DF cases. Poisson regression model is a standard model for multivariate analysis with count variables [55]. Compared with the general Poisson regression model, the geographically weighted Poisson regression (GWPR) not only quantifies the count variables with spatial location attributes, but also detects non-stationary relationships between the variables. By examining the spatial distributions of the coefficients, we can gain a better understanding of the spatial heterogeneity of the spatial processes [56]. Therefore, GWPR is considered as a suitable model for this research. The following was the equation: (3)$$\ln\left(A_{i}\right)=\ln\left(P_{i}\right)+\beta_{0}\left(u_{i},v_{i}\right)+\sum_{k=1}^{p}\beta_{k}\left(u_{i},v_{i}\right)X_{ik}+\varepsilon .$$

In the equation, $A_{i}$ is the amount of DF cases of *Jiedao*
i; $P_{i}$ is the population at risk of *Jiedao*
i; $\left(u_{i},v_{i}\right)$ is the coordinate of *Jiedao*
i; $\beta_{k}$ is the coefficient of the *Jiedao k*. explanatory variable in *Jiedao*
i, and it is a function changed by location.

In this study, five models are constructed by GWPR in order to test whether and to what extent the commuting flows from/to epidemic areas affect the spread of DF. Model 1 only considers four control variables. Model 2 includes FlowIn besides the four control variables. Model 3 replaces FlowIn with FlowOut. Models 4 and 5 include EpiFlowIn or EpiFlowOut besides the four control variables. At last, analysis of variance (ANOVA) is used to examine whether the environment of epidemic areas could explain the spatial variation of the commuting flows from/to epidemic areas.

## 3. Results

### 3.1. Exploratory Analysis

The distribution of DF in Guangzhou in 2014 is shown in Figure 2. The DF in Guangzhou mainly cluster in the conjunction of Baiyun, Tianhe, Yuexiu, Liwan, and Haizhu. The value of the global Moran’s I test for DF is 0.17 with *p* < 0.001, indicating that there is a significant spatial clustering of DF in Guangzhou. Then epidemic areas (high-high clusters with *p* < 0.001) are detected by the local Moran’s I and are shown in red in Figure 2b.

Table 2 shows the descriptive statistics of the variables used in the regression models. The number of DF cases vary from 0 to 1445, with a mean value of 255 and standard deviance of 258. Commuting flows vary greatly among different *Jiedaos.* The spatial distributions of commuting flows are mapped in Figure 3. The darker the color, the larger the commuting flows. FlowIn (Figure 3a) and FlowOut (Figure 3b) have a very similar pattern. There are several high general commuting flow areas in the north part, in the central part, and in the south part (Figure 3a,b). While compared to commuting flow, epidemic flows are mainly concentrated in the surrounding areas of the epidemic area (Figure 3c,d).

### 3.2. GWPR Results

Before GWPR, four control variables and the commuting flow are put into Poisson regression. The result of Poisson regression shows that all the variables are significant with *p* < 0.05 and the Moran’s I test of residuals is nearly 0.2 with *p* < 0.001, indicating that there is a significant spatial clustering and GWPR is suitable. AIC (Akaike Information Criterion) and percent deviance explained are considered as two important indicators for GWPR model fitness. A model with a lower AIC and a higher percent deviance explained is considered to be better. Table 3 compares the model fitness of the five models.

First, the null model (Model 1) that only contains the control variables is applied to examine to what extent DF can be explained only by the local environments. Then, the commuting flows are considered into Model 2 and Model 3 in addition to the control variables. The result shows that the percent deviance explained increases by 3% from 0.67 to 0.7, and the AIC declines by 752/900 from 7781 to 7029/6881. After the commuting flows from/to the epidemic areas were included into Model 4 and Model 5, the percent deviance explained increases by 4% or 5% from 0.7 to 0.74/0.75, and the AIC reduced more (from 7029/6881 to 6066/5864). The results of Model 4 and Model 5 demonstrate a better model fitness, indicating that the commuting flows from/to epidemic areas has a better explanatory power in DF transmission than general commuting flows. The results of the five models answer Question 1.

The impacts of the commuting flows from/to epidemic areas on DF are summarized in Table 4 and mapped in Figure 4. Table 4 shows that the mean value of standardized coefficient for Epiflowin is 0.647 and the median value is 0.727, which is greater than the other coefficients (local_popdens, local_rioldhous, local_NDVI, local_riowater). Similarly, the mean value of the standardized coefficient for EpiFlowout is 0.977, and the median value is 0.842, which is far greater than the other coefficients. This again demonstrates that the commuting flows from/to epidemic *Jiedao* play an extremely important role in explaining DF outbreak (Question 1). Figure 4 shows that the impacts of commuting epidemic flows on DF are not homogeneous; they vary in space. However, the effects are stronger in the south and weaker in the north. This finding answers Question 2.

Why does the effect vary in space? Since the local environmental factors have been controlled, the main reason for the strong or weak effect may be related to two types of epidemic environments (*Epi_env_from* and *Epi_env_to*). To test this assumption, the coefficients of EpiFlowIn and EpiFlowOut are classified into four groups by geometrical interval in ArcGIS 10.3 (Esri, RedLands, CA, USA ) (Figure 4) and ANOVA is applied to find the difference of two types of epidemic environments among four groups. The analysis result is provided in the following section. 

### 3.3. ANOVA Results

The mean *Epi_env_from* value of each group is shown in Table 5. Group 3 of EpiFlowIn has the strongest significant effect (1.219), while Group 1 has the weakest significant effect (0.144). Among the four groups, Group 3 ranks first in Epi_liv_popdens, Epi_liv_rioldhous, and Epi_liv_riowater, while it ranks last in Epi_liv_NDVI, indicating the inbound commuters of Group 3 who live in highly populated epidemic areas with poor housing conditions and more water surfaces and less vegetation. Obviously, the inbound commuters of Group 3 are more likely to be infected at home; thus, it is easy for them to bring the virus out when they go to work.

The mean *Epi_env_to* value of each group is shown in Table 6. A very similar pattern is observed when compared to Table 5. Among four groups, Group 3 have the strongest effect (1.829), while Group 1 has the weakest significant effect (0.0606). The local commuters of Group 3 who work in highly populated epidemic areas with poor housing conditions and more water surfaces and less vegetation are more likely to be infected at the work place and bring the virus back home. The result of Table 5 and Table 6 answer Question 3.

## 4. Discussion

Human mobility and environment have been considered as two important factors that drive the outbreak and transmission of DF. Since commuting is considered to be the main travel for urban residents, commuters provide many opportunities for the spread of infectious diseases via mosquitos’ contact. This study examines the effect of the environment and commuting flows from/to epidemic areas on the spread of DF in 2014 at the *Jiedao* level in Guangzhou by using GWPR and ANOVA. 

The analytical results suggest that intra-urban commuting flows from/to epidemic *Jiedaos* have a significant impact on DF diffusion and have a better explanatory power than general commuting flows in DF diffusion. An unexpected finding is that the effect of commuting flows from/to epidemic areas was much greater than the local environmental indicators such as population density, old housing conditions, NDVI, and water areas (Table 4). This finding suggests that intra-urban commuting flows from/to epidemic *Jiedaos* would be the main source of DF diffusion. Similar findings were also found in a few previous studies. One is Wen’s study [18], where the author compared commuting and non-commuting DF cases in Tainan City of Taiwan and found that commuting was identified as a significant risk factor contributing to epidemic diffusion. Another recent study conducted by Huang [57] used weekday commuting network to construct an algorithm to analyze the diffusion of two infectious diseases in Taiwan, and the result suggested the availability of the commuting network in predicting epidemic diseases. Besides, Rajarethinam [17] conducted a study about Zika epidemic in Singapore using mobile phone data and found that there were higher odds of Zika cases being reported in the areas that were visited by people from epidemic clusters. 

Besides the effects of commuting flows from/to epidemic areas, this study demonstrates that the effects vary in space and comes up with the reasons for spatial heterogeneity by using ANOVA. The results suggest that the environments of the related epidemic areas contribute to infection diffusion through the move of the commuters. The results show that the *Jiedaos* whose commuters commuted to/from the epidemic areas that have higher population densities, older buildings, more water surfaces, and less vegetation coverage tend to be more influenced by the commuting flows than the *Jiedaos* whose commuters commuted to/from the epidemic areas with better conditions. Commuters in environments where mosquitos easily breed are more likely to be infected and bring the virus out when they commute to work or back home. Commuting flow is similar to a “bridge” linking other places to a local place and facilitating the interaction between places. The results suggest that the environment of other places in the “bridge”, especially epidemic *Jiedaos*, could be an important cause of local infection. In contrast, previous studies mainly focused on the local environment [11,12,13,39,40,41] but neglected environments of the places, especially epidemic areas that commuters commuted from/to. This study bridges the gap and suggests that the living and working environment of commuters contribute to the source of infection diffusion. One recent study also demonstrates that an individual may often be infected at their home or other places where they spend significant amounts of time, such as their place of work, and homes and work places were suggested as possible transmission sources [58]. This finding suggests that the health management officers should pay attention to commute-related environments when combating a DF outbreak in a local place. 

In a word, the findings from this study show us more insights into DF spread and help us better understand the transmission pattern of the disease, which provides useful guidance for targeted control strategies. In the past, DF control mainly relied on mosquito elimination and environmental management. Once an outbreak occurs, large-scale larval habitat elimination or campaigns using administrative boundaries as spatial units were carried out. This approach is time-consuming and less effective. Our results provide targeted strategies of interventions. For example, we would suggest that the spatial targeting should focus on the living and working environments of commuters. This helps us narrow the scope of the control and improve the effective control under limited resources for future outbreaks both in China and elsewhere.

However, this study has some limitations. First, meteorological indicators such as temperature, precipitation, and humidity were not included in this study. However, at a city scale (intra-urban), the meteorological difference among *Jidaos* was little and could be ignored. Second, the time when the mobile phone data were collected is not consistent with the onset of DF. We assumed that human daily commuting patterns do not change much from 2014 to 2016. Third, the spatial scale chosen at the *Jiedao* level is due to the limit of data availability; a finer-grained dataset may be expected in the future research. Last, the behavior of mosquitos was ignored, and mosquitos’ density was not included in the analysis. There were few mosquito monitoring sites in Guangzhou in 2014. We assumed that vector density was even and the exposure to mosquitos was the same in each area of the city. In fact, even if human mobility was high, the probability of DF transmission caused by population mobility will be greatly reduced if the exposure to mosqutios was low or the density of mosqutios was low. In future studies, more risk factors and other routine activity places such as shopping or leisure places were expected to be taken into account in order for better understanding the spread of DF.

## 5. Conclusions

This study illustrates the impact of intra-urban daily commuting flow, especially those from/to epidemic areas, on DF transmission. The results suggest that (1) commuting flows from/to epidemic areas affect the transmission of DF; (2) such effects vary in space; and (3) the spatial variation of the effects can be explained by the environments of the epidemic areas that commuters commuted from/to. Health management officers should pay attention to commute-related environments when combating a DF outbreak in a local place. These findings have important policy implications for making effective intervention strategies, especially when resources are limited.

## Figures and Tables

**Figure 1 ijerph-16-05013-f001:**
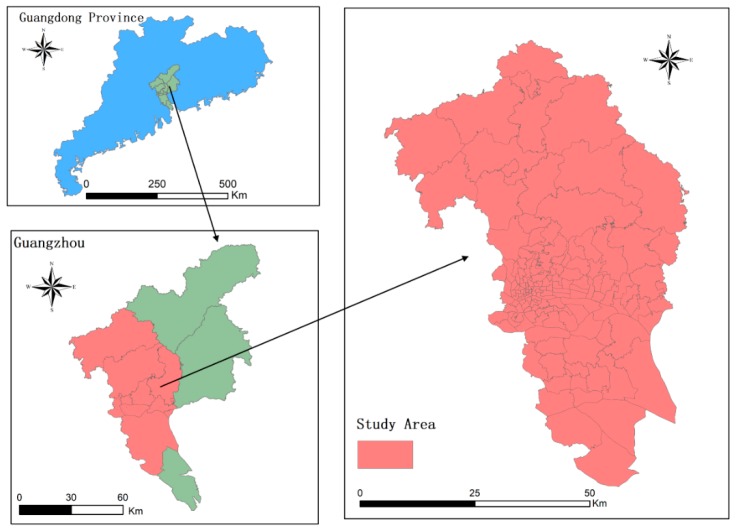
Study area.

**Figure 2 ijerph-16-05013-f002:**
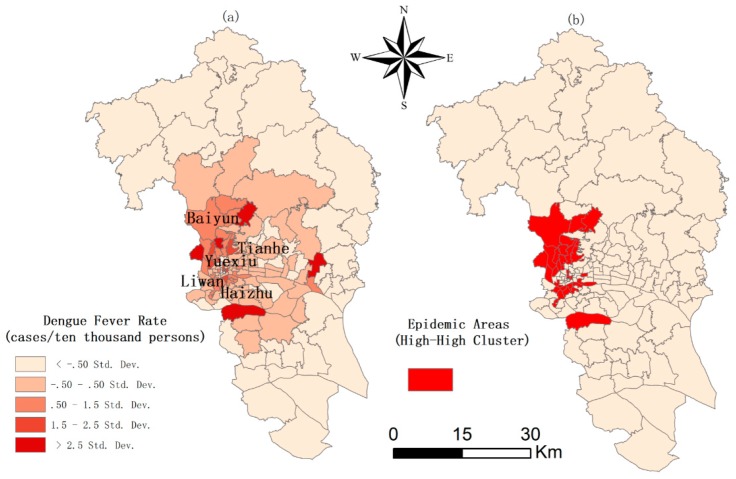
Distribution and cluster analysis of dengue fever in Guangzhou in 2014. (**a**) The spatial distribution of the dengue fever rate; (**b**) epidemic areas identified by local Moran’s I.

**Figure 3 ijerph-16-05013-f003:**
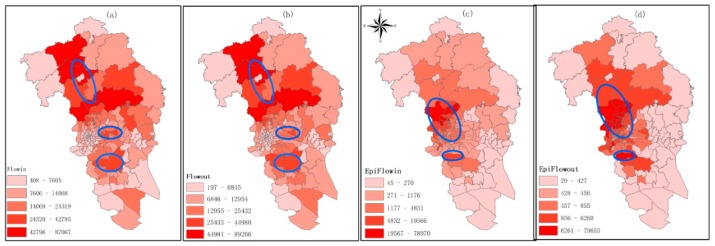
The spatial distribution of commuting flow. (**a**) Spatial distribution of FlowIn; (**b**) Spatial distribution of FlowOut; (**c**) Spatial distribution of EpiFlowIn; (**d**) Spatial distribution of EpiFlowOut.

**Figure 4 ijerph-16-05013-f004:**
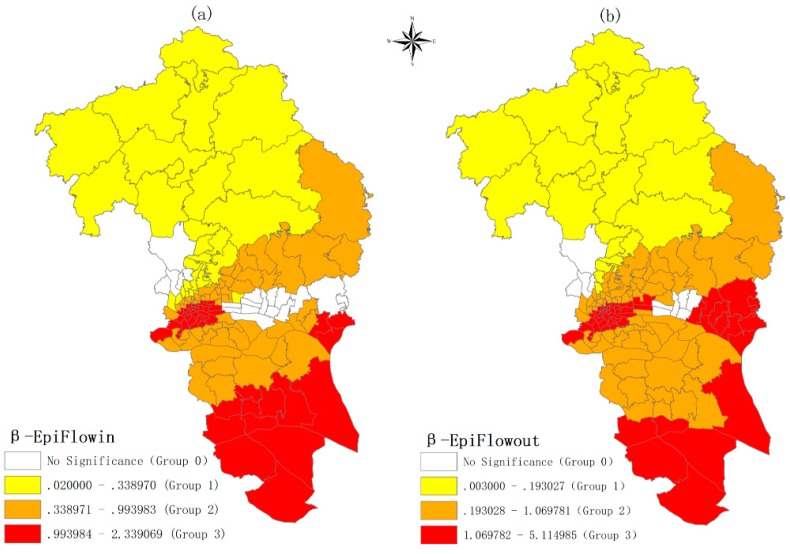
The spatial distribution of coefficient of commuting flow from/to epidemic areas. (**a**) The coefficient of commuting flow from epidemic areas; (**b**) The coefficient of commuting flow to epidemic areas.

**Table 1 ijerph-16-05013-t001:** Detailed description of variables.

Categories	Variables	Description
Commuting Flow		Daily commuting flow
	FlowIn	The total numbers of commuters from other *Jiedao*s
	FlowOut	The total numbers of commuters to other *Jiedao*s
	EpiFlowIn	The total numbers of commuters from epidemic *Jiedao*s
	EpiFlowOut	The total numbers of commuters to epidemic *Jiedao*s
Local environment		*Jiedao*’s local environment
	Local_popdens	*Jiedao*’s population density
	Local_rioldhous	*Jiedao*’s proportion of old houses (built before the 1990s)
	Local_NDVI	*Jiedao*’s normalized difference vegetation index
	Local_riowater	*Jiedao*’s proportion of water areas
Epidemic environment		The environment of the epidemic *Jiedao*s that interact with this local place through commuters
	Epi_env_from	The environment of the epidemic *Jiedao*s from which commuters came (or “come-from” epidemic *Jiedao*s)
	Epi_popdens_from	The population density of the “*come-from*” epidemic *Jiedao*s
	Epi_rioldhous_from	The proportion of old houses of the “*come-from*” epidemic *Jiedao*s
	Epi_NDVI_from	The normalized difference vegetation index of the “*come-from*” epidemic *Jiedao*s
	Epi_riowater_from	The proportion of water areas of the “*come-from*” epidemic *Jiedao*s
	Epi_env_to	The environment of epidemic *Jiedao*s to which commuters traveled (or “go-to” epidemic *Jiedao*s)
	Epi_popdens_to	The population density of the “*go-to*” epidemic *Jiedao*s
	Epi_rioldhous_to	The proportion of old houses of the “*go-to*” epidemic *Jiedao*s
	Epi_NDVI_to	The normalized difference vegetation index of the “*go-to*” epidemic *Jiedao*s
	Epi_riowater_to	The proportion of water areas of the “*go-to*” epidemic *Jiedao*s

**Table 2 ijerph-16-05013-t002:** Descriptive statistics of variables. DF: dengue fever.

Categories	Variable	Min	Max	Mean	Std
Dependent variable	DF cases	0	1445	255	258
Independent variables	FlowIn	408	87,067	13,391	13,535
	FlowOut	197	89,266	13,391	14,293
	EpiFlowIn	45	78,970	3038	8198
	EpiFlowOut	29	79,653	2867	8398
Control variables	local_popdens (10,000 people/km^2^)	0.01	8.43	1.96	2.09
	local_rioldhous (%)	0.71	80.19	29.20	21.48
	local_NDVI	0.036	0.284	0.123	0.556
	local_riowater (%)	0.00	46.25	6.79	8.83

**Table 3 ijerph-16-05013-t003:** Geographically weighted Poisson regression (GWPR) model results.

Models	Variables	Percent Deviance Explained	AIC
**Model 1**	Four control variables	0.6702	7781.21
**Model 2**	Four control variables, FlowIn	0.7028	7029.15
**Model 3**	Four control variables, FlowOut	0.7091	6881.03
**Model 4**	Four control variables, EpiFlowIn	0.7437	6066.15
**Model 5**	Four control variables, EpiFlowOut	0.7523	5864.58

Notes: Four control variables include: local_popdens, local_rioldhous, local_NDVI, local_riowater.

**Table 4 ijerph-16-05013-t004:** The coefficients of Model 4 and Model 5.

Models	Variable	Mean	Median	Min	Max	Std
Model 4	Pop_density	0.0695	0.0336	−0.7449	1.1678	0.3648
	Rio_oldhouse	0.0727	0.1905	−1.4327	0.5583	0.3500
	NDVI	−0.0807	−0.1155	−0.6141	0.5604	0.1978
	Rio_water	0.0925	0.0724	−0.2849	1.4412	0.2249
	EpiFlowin	0.6479	0.7273	−0.3482	2.3390	0.5005
Model 5	Pop_density	0.0564	0.0361	−0.7548	1.1754	0.3696
	Rio_oldhouse	0.1027	0.2267	−1.4238	0.5921	0.3540
	NDVI	−0.0838	−0.0944	−0.6153	0.5648	0.2001
	Rio_water	0.0828	0.0553	−0.2789	1.4033	0.2218
	EpiFlowout	0.9775	0.8430	−0.0385	5.1149	0.8904

**Table 5 ijerph-16-05013-t005:** The mean value of “Epi_env_from” of each group.

Group	β-EpiFlowIn	Epi_popdens_from	Epi_rioldhous_from	Epi_NDVI_from	Epi_riowater_from
0	No significance	1.6965 *	18.1868 *	0.0993 *	7.8068 *
1	0.144	1.6510 *	18.9830 *	0.1053 *	4.8462 *
2	0.7083	1.8057 *	23.1628 *	0.0975 *	8.7734 *
3	1.2198	2.3743	32.655	0.0885	10.728

* The mean difference of each group compared with Group 3 is significant at the 0.05 level.

**Table 6 ijerph-16-05013-t006:** The mean value of “Epi_env_to” for each group.

Group	β-EpiFlowout	Epi_popdens_to	Epi_rioldhous_to	Epi_NDVI_to	Epi_riowater_to
0	Nosignificance	1.9231 *	7.7114 *	0.0973 *	7.8192
1	0.0606	1.6194 *	22.3415 *	0.1037 *	3.6239 *
2	0.7232	2.1908 *	27.0545 *	0.0947 *	8.4258
3	1.8289	2.7388	39.1045	0.0866	10.3737

* The mean difference of each group compared with Group 3 is significant at the 0.05 level.

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
