# Peer review of "Examining the Effect of the Environment and Commuting Flow from/to Epidemic Areas on the Spread of Dengue Fever"

_ijerph, 2019, doi:10.3390/ijerph16245013_

Round 1
Reviewer 1 Report
The paper is wellorganized and I am quite interested in the topic. This paper built a bridge between commuting behavior study and space epidemiology. I really like the paper as they provide a space-time behavior perspective on DF studies and use the big data to investigate the relationship. But, I would like to provide some minor questions and comments additionally.
Strength:
The paper uses the mobile data to study the impact of commuting flow on DF. And they study both the out-flow and in-flow effects. GWPR models are used to investigate the effect of commuting flow considering spatial heterogenous.Improvement required
The authors should discuss about the spatial scale they used in this study. Actually, jiedao is relatively large for the study of DF transformation. Some discuss should be made in the limitation part. The authors only have one-day mobile data. Thus some travelers or those conduct non-commuting trips may be identified as commuters using their methods. Please provide the proportion of commuters identified and mage a comparison with existing studies to show the representativeness. For the model, I wonder if commuting flow may be highly related to population density, especially for the out-flow.Author Response
Please see the attachment.

Reviewer 2 Report
It is an interesting study to analyze the influence of commuters on the outbreak of DF. It must be useful to develop the understanding of DF spread.
However, I did not understand why the authors analyzed the data by GWPR. If there is a mechanism that the occurrence of DF varies by locations, the analysis by GWPR is justifiable.
I recommend the authors to add explanation why the phenomenon of DF is supposed to have spatial heterogeneity.
I add some minor comments below.
Section 3.2
The use of AICc for Poisson regression is not adequate, because the derivation of AICc depends on the assumption that the model has normally-distributed residuals. AIC should be used instead, as it is asymptotically correct for Poisson regression.
Line 219
The first letter of "Poisson" should be a capital letter.
Line 220
"passion" must be "Poisson."
